# Creep and Residual Properties of Filament-Wound Composite Rings under Radial Compression in Harsh Environments

**DOI:** 10.3390/polym13010033

**Published:** 2020-12-23

**Authors:** Frederico Eggers, José Humberto S. Almeida Jr, Tales V. Lisbôa, Sandro C. Amico

**Affiliations:** 1SENAI Institute of Innovation in Polymers Engineering, São Leopoldo 93025-753, Brazil; fredericoeggers@yahoo.com.br; 2Department of Mechanical Engineering, Aalto University, 02150 Espoo, Finland; 3Mechanics and Composite Materials Department, Leibniz Institute of Polymer Research, 01069 Dresden, Germany; tales-lisboa@ipfdd.de; 4PPGE3M, Federal University of Rio Grande do Sul, Porto Alegre 91501-970, Brazil; amico@ufrgs.br

**Keywords:** creep, filament-wound ring, hygrothermal conditioning, radial compression

## Abstract

This work focuses on the viscoelastic response of carbon/epoxy filament-wound composite rings under radial compressive loading in harsh environments. The composites are exposed to three hygro-thermo-mechanical conditions: (i) pure mechanical loading, (ii) mechanical loading in a wet environment and (iii) mechanical loading under hygrothermal conditioning at 40 ${}^{\circ }$C. Dedicated equipment was built to carry out the creep experiments. Quasi-static mechanical tests are performed before and after creep tests to evaluate the residual properties of the rings. The samples are tested in (i) radial compression, (ii) axial compression, and (iii) hoop tensile strength. Different laminates wound at off-axis orientations are manufactured via filament winding and analyzed. Key results show that creep displacement is affected by both hygrothermal and mechanical conditionings, especially at a higher temperature. Moreover, residual properties are quantified showing that creep generates permanent damage in the cylinders.

## 1. Introduction

Carbon fiber-reinforced polymer (CFRP) composite structures have been increasingly used in many engineering applications given their high specific stiffness and strength and high corrosion resistance [1,2,3]. Tubes and pipelines produced by filament winding (FW) are often used in fluid transportation and this environment can change their mechanical behavior [4]. If so, the structure is continuously and simultaneously under thermal, environmental, and mechanical stresses, therefore durability and hygrothermal aspects should be accounted for in the design of such components. When the structure is under wet conditions and high temperature, the long-term behavior of a CFRP may be affected by physical (e.g., changes in glass transition temperature—$T_{g}$) and chemical aging (e.g., changes in molecular weight, oxidation). Some of the hygrothermal aging effects are reversible, such as the effect on $T_{g}$, and others depend upon the material system and environmental condition, such as the plasticization of the matrix [5].

Swelling, plasticization, slow hydrolysis of the polymeric matrix, and slow attack of the fluid to the fiber/matrix interface may yield the loss of properties, influencing mechanical behavior and, therefore, the durability of the structure. Given that, understanding the long-term response of a CFRP structure under thermomechanical load and harsh environment is especially important when it represents the actual loading scenario of the structure [6,7].

This long-term behavior known as creep refers to a time-dependent deformation under a constant load and temperature [8]. For polymer composites, creep resistance is directly associated with viscoelastic strain and fiber/matrix interfacial behavior [9], and they exhibit a time-dependent degradation in modulus (creep or stress relaxation) and strength (creep-rupture) associated with the viscoelasticity of the polymer matrix [10].

Merah et al. [11] studied the effect of harsh weather and seawater on glass fiber-reinforced epoxy composites focusing on residual properties of tubes in outdoor environment. The rings were exposed to seawater for up to 10,000 h and it was observed a reduction in both tensile strength and fracture strain after 300 h. Farshad and Necola [12] studied the effect of environmental conditioning on the long-term behavior of glass/polyester tubes under water at room temperature. The experimental data were obtained over 2 years and extrapolated to up to 50 years. The results showed that strength in radial compression of the wet tubes after 1000 h was reduced to ≈60% of the dry tube strength. Guedes et al. [13] performed creep tests in glass/polyester tubes and determined both short and long-term rupture energies. They concluded that pre-conditioning in water at 50 ${}^{\circ }$C reduced the initial stiffness and strength in ≈4% and ≈60%, respectively, considering a 50-year lifetime. Yang et al. [14] evaluated the creep behavior of CFRP tubes under bending using the time-temperature-stress superposition (TTSSP) principle and Findley model. Creep tests were carried out at stress levels of 45–75% of the ultimate bending strength at constant temperatures ranging from −60 ${}^{\circ }$C to 100 ${}^{\circ }$C for 500 h. The creep rate increased with increasing stress and temperature levels, whereas long-term deformation did not exceed the ultimate bending strain. Keller et al. [15] evaluated moisture effects on thermal and creep behavior of woven carbon/epoxy cylinders. They observed that samples immersed in water (for up to 18 months) presented greater creep compared to the dry ones (ambient conditions).

Considering that CFRP composite tubes are very stiff under normal operating conditions, their viscoelastic behavior is usually disregarded in structural designs. However, in structural applications, composite tubes and pipelines are wound at off-axis angles, therefore, there is a stronger contribution of the polymeric matrix and viscoelasticity cannot be disregarded, since it can possibly lead to premature matrix cracking and weakening of the fiber/matrix interface.

This study aims at evaluating the creep response of carbon/epoxy filament-wound rings under three different hygrothermal and mechanical scenarios with a comprehensive experimental campaign. Water uptake and post-creep residual properties are determined, and the influence of the winding sequence is studied.

## 2. Materials and Methods

### 2.1. Materials and Manufacturing

A carbon/epoxy towpreg from TCR Composites, comprised of a Toray T700-12K-50C carbon fiber reinforcement and UF3369 epoxy resin matrix with a fiber volume fraction ($V_{f}$) of 72% [5], is used. The tubes are manufactured with a filament winding machine equipped with a KUKA 140 L100 robot. Design of the laminates is carried out with CadWind software [16].

Cylindrical tubes are manufactured onto a stainless-steel mandrel with 50.8 mm inner diameter and 1000 mm in length. The following tubes are manufactured: [±60], [±75], [±90], [±60/±90], and [±75/±90]. The hoop layer is here defined as [±90]; however, the actual angle is ±89.6${}^{\circ }$. The winding patterns for the [±60] and [±75] families are 1/1, which means one diamond around the circumference. The winding pattern is generated in helical and polar windings, but no in hoop winding.

The cylinders are cured in an oven with air circulation for 5 h at 120 ${}^{\circ }$C [17]. Then, the system is cooled down to room temperature and the composite tube is removed from the mandrel. The tubes are cut off into 50-mm long rings. A diamond saw aided by water is used for cutting, and the samples are later polished.

### 2.2. Design of the Creep Equipment

A dedicated creep testing equipment is designed and built for carrying out this investigation. It consists of a sliding platform mounted inside a rectangular metallic structure (Figure 1a–d). Linear bearing housings are mounted at the corners of the platform and attached to vertical guides. The function of the bearings is to restrict any lateral movement, allow vertical displacement, and reduce friction between the platform and the guides.

The equipment is manufactured with square carbon steel profiles, with bottom plate and platform in SAE 1020 carbon steel. The vertical guides are machined with SAE 1045 carbon steel bars. To allow liquid storage, the sides are sealed with polycarbonate plates (Figure 1c). The structure is subjected to surface treatments and electrostatic painting and the guides are covered with chromium for corrosion protection.

To monitor the displacement throughout the test, an instrumentation system is assembled (Figure 1b,c). Tubular immersion electrical resistance (1500 W), K-type thermoresistance, Novus N480D digital temperature controller, Balluff micropulse linear variable differential transformer (LVDT), Novus Logbox AA data logger, 24-V power supply, and NHS no-break compose this system. The LVDT sensor is fixed to the equipment and the magnet is bolted to the moving platform to capture platform displacement (Figure 1c). When the platform travels, a signal is processed and stored (at 0.05 Hz) by the datalogger signal recorder, which transforms it into displacement measurement and allows instantaneous visualization on the computer screen via the LogChart II software.

For creep tests under hygrothermal conditions, the water temperature is controlled by a digital controller connected to the thermoresistance. To control the amount of water inside the chamber, a level sensor is assembled along with a pump (Figure 1c). Water circulation in a closed circuit is obtained with another pump. A 24 V power supply is used, and a no-break system provides an uninterrupted supply of power throughout the test.

Given the large number of samples to be tested in creep, complimentary simpler equipment is built to speed up the tests. This device, nevertheless, is only able to carry out unconditioned creep, i.e., mechanical loading without hygrothermal conditioning (see Figure 1d).

### 2.3. Water Uptake Determination

To determine water uptake of the rings, the specimens are immersed in stainless-steel containers with distilled water at room temperature (23 ± 2 ${}^{\circ }$C) or 40 ± 2 ${}^{\circ }$C kept inside an oven. These are the same temperatures as the creep tests. Before conditioning, the samples are dried in an oven at 100 ${}^{\circ }$C for 24 h to remove initial humidity and weighed to obtain a reference value. They are then weighed again and placed in the oven for an extra 3 h. These steps are repeated until the samples reach equilibrium. The dried samples are immersed in distilled water at predetermined temperature conditions for 45 consecutive days (1080 h). Three samples for each laminate are monitored daily. The water uptake (*M*) is calculated using $M=\left(M_{f}-M_{0}\right)/M_{0}$, where $M_{f}$ and $M_{0}$ are the wet and dry masses, respectively.

Fick’s analytical model [5,18] (Equation (Equation 1)) is used to analyze the experimental data as
(1)$$M_{t}=M_{\infty }\left\{1-\exp\left[-7.3{\left(\frac{Dt}{h^{2}}\right)}^{0.75}\right]\right\}\;,$$
where $M_{t}$ corresponds to the water uptake at a particular time *t*, $M_{\infty }$ defines the mass at a quasi-equilibrium state, *D* is the diffusion coefficient, and *h* is the specimen thickness. The diffusion coefficient is calculated from the absorption curve, as follows
(2)$$D=\pi {\left(\frac{h}{4M_{\infty }}\right)}^{2}{\left(\frac{M_{j}-M_{i}}{\sqrt{t_{j}}-\sqrt{t_{i}}}\right)}^{2}\;,$$
in which the subscripts in *M* and *t* terms refer to a particular mass and time, respectively. It is worth mentioning that $t_{i}$ and $t_{j}$, must be in the transient regime, in which water uptake increases linearly with the square root of time. The slope of this straight line determines diffusivity, or water diffusion, *D*,[19]]. In this regime, the water is heterogeneously distributed through-the-thickness and therefore water uptake increases until saturation is reached, where the specimen has a mass $M_{l}$ at a time $t_{l}$. It is important to highlight that (i) *i* and *j* must be lower than *l*, and (ii) *i* and *j* must be higher than 0. The *D* cannot be measured in the pseudo-equilibrium regime because after the mass *M* reaches a plateau with mass $M_{\infty }$, the water is homogeneously distributed throughout the specimen thickness.

### 2.4. Thermal and Dynamical-Mechanical Analysis

To determine the degree of cure of epoxy, differential scanning calorimetry (DSC) analysis is performed. A DSC Q20 TA Instruments analyzer is used, with nitrogen with 50 mL/min, and a heating ramp of 10 ${}^{\circ }$C/min up to 250 ${}^{\circ }$C. The uncured towpreg, i.e., which has not yet undergone curing, was analyzed in comparison with two other samples, a regularly cured non-conditioned ring, and a cured conditioned (hot water at 40 ${}^{\circ }$C for 240 h) ring.

Dynamic mechanical analysis (DMA) is performed in a DMA 2980 equipment from TA Instruments to determine glass transition temperature ($T_{g}$), storage ($E^{\prime }$) and loss ($E^{\prime \prime }$) moduli, and loss factor (tanδ). Samples with prismatic geometries are obtained from the rings [±75] (see Table 1) being: (i) unconditioned, (ii) conditioned in water at room temperature for 240 h, and (iii) conditioned in water at 40 ${}^{\circ }$C for 240 h. The samples are tested using a double cantilever beam mode with a frequency of 1 Hz and an amplitude of 15 μm. The $T_{g}$ is determined as the intersection between the extrapolation of the elastomeric plateau and the glass region of the storage modulus ($T_{\left(g,E^{\prime }\right)}$), as the temperature corresponding to the loss module peak ($T_{\left(g,E^{\prime \prime }\right)}$), and as the temperature of the tan delta peak ($T_{\left(g,\tan\delta \right)}$).

### 2.5. Creep: Testing and Modeling

Initial quasi-static radial compressive tests (Figure 2a) are performed in an Instron Universal testing machine model 3382 (load cell of 100 kN) following the recommendations of ASTM D2412-11 standard, at a speed of 2.5 mm/min. The experimental test concerns the determination of the external load-deflection curve. Five samples are tested to obtain “reference” non-aged composite properties. Ring stiffness (RS), percentage ring deflection quota (*P*), and stiffness factor (SF) are calculated as follows [17].
(3)$$RS=\frac{F}{\Delta_{y}}{\left(1+\frac{\Delta_{y}}{2d}\right)}^{3}\;,$$
(4)$$P=\frac{\Delta_{y}}{d}\times 100\%\;,$$
(5)$$SF=EI=0.149r^{3}\times RS\;,$$
where *F* is the applied load, *d* and $\Delta_{y}$ correspond to the outer diameter of the specimen and its change in the load direction, respectively, *E* is the bending modulus and *I* is the tube’s inertia. The parameter *r* is the mid-wall radius, obtained by subtracting wall thickness from the outside diameter and dividing it by two. Since the lateral displacement of the ring can be much larger than the tube thickness ($\Delta_{y}/t>>1$, where *t* corresponds to the wall thickness), changes in the radius of curvature are expected. Moreover, the greater the deflection at which RS or SF is determined, the greater the magnitude of the deviation from the true value. Assuming that the ring deformed shape remains elliptic during the loading application, a correction factor ($C={\left(1+\Delta_{y}/2d\right)}^{3}$) is applied to RS, and, therefore, to SF. Furthermore, following Almeida Jr. et al. [17], the outside diameter is used instead of the change in inside diameter, since the samples have different wall thicknesses.

Creep tests are performed under constant loading (Figure 1) corresponding to 25% of the maximum compressive load of each type of laminate. Five samples are simultaneously subjected to radial compression under the same loading for 240 h (10 days). This period is long enough to allow matrix saturation, but avoiding tertiary creep which is outside the scope of this work. Three scenarios are studied: (i) only mechanical loading; (ii) simultaneous mechanical loading and water immersion at room temperature, and (iii) simultaneous mechanical loading and water immersion at 40 ${}^{\circ }$C. Recommendations of the European standard EN 1227:1998, used to determine long-term ultimate relative ring deflection under wet and hot conditions, are herein generally followed. However, although the standard suggests 50 ${}^{\circ }$C of temperature, it is quite difficult to keep this temperature throughout the creep test [8,20]. Based on several pre-tests, 40 ${}^{\circ }$C was found the maximum temperature that allowed steady conditions considering the available system. Table 1 presents the adopted nomenclature for all creep conditions considered in this study.

Several analytical models have been employed in attempt to predict the viscoelastic response of polymers and polymer composites, including Findley [2], Burgers [9], and Kohlrausch-Williams-Watts (KWW) [21]. Findley’s is perhaps the most widely used model for fiber-reinforced polymer composites given the good balance between accuracy and easy implementation. The Findley law is here defined as follows:(6)$$d\left(t\right)-d_{0}=At^{n}$$
where d(t) corresponds to the displacement at time *t*, $d_{0}$ to the instantaneous displacement (or instantaneous creep compliance), *A* to the amplitude of the transient creep compliance and *n* is a stress-independent material constant dependent (usually n<1) [22]. Equation (Equation 6) differs from the original Findley’s law, which considers creep strain instead of displacement. The “hardening” here is still taken into consideration through the constant *n*, i.e., the hardening is only time-dependent.

### 2.6. Post-Creep Residual Properties

To obtain residual properties, the post-creep rings (five samples of each type) are subjected to quasi-static radial compression (Figure 2a), axial compression (Figure 2b), and hoop tensile tests (Figure 2c). The conditions of these tests as well as the results (e.g., load versus displacement curves) are presented in detail in Ref. [23]. The notches for hoop tensile test specimens (Figure 2c) are machined after creep tests to ensure a fair comparison and proper residual properties. Briefly, all rings subjected to creep tests have the same geometrical characteristics, therefore the notches are machined after the creep tests.

## 3. Results and Discussion

### 3.1. Towpreg Curing Investigation

Figure 3 presents Differential scanning calorimetry (DSC) results for uncured towpreg and cured composites at room temperature and after immersion in hot water at 40 ${}^{\circ }$C for 240 h. For the uncured towpreg (1st heating), the curing process occurs with a typical heat release, thus generating a well-defined exothermic peak at ca. 165 ${}^{\circ }$C. For the cured towpreg (2nd heating), that peak disappears and the $T_{g}$ is identified as 97 ${}^{\circ }$C. Comparing the DSC curves of the towpreg (2nd heating) and the unconditioned composite, an adequate degree of curing of the epoxy resin can be expected since the curves coincide, and no exothermic peak is observed.

For the wet-conditioned composite, the $T_{g}$ decreases to 93 ${}^{\circ }$C, there is a drop in heat flow and the curve does not stabilize. This is caused by the presence of absorbed moisture from the immersion in water. De’Nève and Shanahan [24] concluded that 1% of water uptake in epoxy resin yields 8 ${}^{\circ }$C of reduction in $T_{g}$. Even in the case of exposure at room temperature, water absorption in the free volume of the epoxy network can increase chain mobility and reduce $T_{g}$[25]. This decrease in $T_{g}$ can lead to an increase in voids and cracks as well as debonding between fiber and matrix [26]. Indeed, water and temperature can substantially affect the initial stiffness and viscoelastic behavior of a composite [27].

Figure 4 shows the storage ($E^{\prime }$) and loss ($E^{\prime \prime }$) moduli, and loss factor (tanδ) curves obtained. In general, the shape of the curves for unconditioned and conditioned situations is similar. From Figure 4a, $E_{max}^{\prime }$ reduces as the intensity of environmental effects increases, mainly attributed to water absorption [5,6,10]. In the glassy state, the aged composites show similar behavior, which is possibly due to post-curing during aging [5]. Regarding loss modulus ($E^{\prime \prime }$), a small increase is noted when the sample is aged at room temperature, whereas when under hot water, the opposite effect is observed. Loss modulus is related to energy dissipation of the material, and structures with weak interfacial bonding tend to dissipate more energy. Hygrothermal conditioned specimens showed lower loss modulus, as can be seen in Figure 4b.

The $T_{g}$ values obtained from the DMA analysis are presented in Table 2. The presence of water reduces the $T_{g}$ of the material and this is more significant under more aggressive conditioning (HW). Comparing ML and ML + WRT conditions based on the different $T_{g}$ determination approaches ($T_{\left(g,E^{\prime }\right)}$, $T_{\left(g,E^{\prime \prime }\right)}$ and $T_{\left(g,\tan\delta \right)}$), the reduction is of 4 ${}^{\circ }$C, 7 ${}^{\circ }$C, and 6 ${}^{\circ }$C, respectively. Comparing ML with ML + HW, the reduction is of 10 ${}^{\circ }$C, 7 ${}^{\circ }$C, and 13 ${}^{\circ }$C, respectively. And comparing the two aged specimens ML + WRT and ML + HW, the reduction is of 6 ${}^{\circ }$C, 0 ${}^{\circ }$C, and 7 ${}^{\circ }$C, respectively. Based on these values, water is found more detrimental to $T_{g}$ than temperature. According to Guen-Geffroy et al. [28] and Yang et al. [14], when epoxy resin is aged in a wet environment, the decrease in its $T_{g}$ can be primarily associated with plasticization. In general, these values are consistent with the values from DSC, and the $T_{\left(g,\tan\delta \right)}$ values are the closest ones.

### 3.2. Water Uptake

Figure 5 displays the water uptake experimental results and fitted curves of all rings. Table 3 presents all input parameters used to feed the Fick model. An initial observation indicates that Fick’s law fits the experimental data very well in both cases (water at room temperature and hot water). It is also noticeable that fiber orientation plays an important role in water uptake. Considering that the rings are not subjected to any mechanical loading during the water absorption tests, it is expected that the higher (i.e., towards hoop direction) the winding angle, the lower the water uptake, since water diffusivity along the fiber direction is greater than in other directions.

In addition, samples with fibers at 90${}^{\circ }$ (hoop-oriented) show the lowest water absorption given that the tows are placed side-by-side with a slight overlap, which decreases the possibility of gaps and, hence, voids, which promote water uptake. In helical windings, a layer is formed after winding of several circuits, forming the so-called cross-over and inter-crossing areas. These areas may generate resin-rich zones due to extra pressure caused by the tow placement, which may then promote an increase in water absorption for helical-wound laminates. Thus, Fick’s model may not fit well all laminates since it does not consider these regular laminate and inter-crossing areas, but considers them regular and homogeneous laminates.

Moreover, as one can see in Figure 5, the temperature strongly influences water uptake. Plasticization and hydrolysis are more pronounced when the samples are under both wet and hot environment, reflecting in a reduction of mechanical properties. It is valid to mention that free volume does not take place since the exposed temperature is far away from the $T_{g}$. However, mention must be given that the kinetics of the diffusion process is dependent on both water uptake and temperature.

Figure 6a,b depicts residual plots to compare water uptake predictions from Fick’s law with experimental data. As can be seen, the deviation between predictions and experiments is low throughout the initial period of time considered, which ensures that Fick’s law predictions, indeed, fit very well the experimental results, except for the sample [±60/±90]. In the case of this sample, there is a fast transient diffusion at the beginning of the water absorption followed by a weight gain at a slower diffusion rate. This characterizes a two-step Fickian diffusion behavior, which cannot be captured by the used Fick’s model. This second stage, which for the other samples represents a pseudo-equilibrium state, still presents a gradual increase in water uptake. The second stage experienced by the [±60/±90] sample is attributed to a relaxation phenomenon, where this water uptake is related to enhanced structural relaxation [29]. This behavior could be captured by including the relaxation rate in Equation (Equation 1).

### 3.3. Creep Performance

The quasi-static results used to calculate both applied creep load residual properties are presented in Ref. [23]. The applied creep load for each case (laminates [±60], [±75], [±90], [±60/±90], and [±75/±90]) and thickness with the standard deviation of the samples are shown in Table 4.

Figure 7 depicts the creep curves for the rings. First, the lower the fiber angle, the higher the deflection, which is expected since as the winding angle approaches the hoop direction, the ring becomes more dependent on the fiber response. And the lower the angle, the more viscoelastic-dependent the structure is due to the greater influence of the polymer matrix (Figure 7a–e). Furthermore, the double-layer laminates deflect less than the single-layer ones, which is also expected given the added stiffness from the extra hoop layer in these rings (Figure 7d,e).

Regarding the effect of the different conditions, the rings under ML + WRT show slightly greater deflection, which can be directly attributed to the water uptake (Figure 5). This difference is very low because the water uptake is also low (less than 0.6% and 2% for the single-walled and double-walled rings, respectively—Figure 5a,b), with limited damage to the fiber/matrix interface. Nonetheless, when creep tests are carried out in hot water (ML + HW), the deflections are much more pronounced. For example, comparing the results after 100 h for ML and ML + HW, the increase in deflection is approximately 39%, 27%, 36%, 44%, and 50% for the [±60], [±75], [±90], [±60/±90] and [±75/±90] laminates, respectively. This behavior, i.e., higher deflection for samples under ML + HW, is expected since creep phenomenon is time- and temperature-dependent, and the effect is more pronounce when the rings are at 40 ${}^{\circ }$C compared to room temperature (23 ${}^{\circ }$C). It is also worth mentioning that none of the rings entered the tertiary creep stage, presenting only a transient time-dependent primary creep followed by a steady-state secondary creep for rings under ML and ML + WRT (i.e., constant strain rate). Even so, the samples under ML + HW show a slightly different secondary creep stage, with higher strain rate, in which the deflection keeps decreasing over time, showing a “quasi-steady state”. Therefore, these rings show a thermally activated glide creep mechanism, typical of viscoelastic materials, such as epoxy. And given their accelerated deformation and ovality, these rings are more likely to enter tertiary creep compared to the samples under ML and ML + WRT.

Creep phenomenon in polymeric systems occurs due to the molecular motion in the epoxy network arrangement, making them susceptible to deform over time [21]. This flow results from the molecular motion of polymeric chains to minimize localized energy [30]. For higher temperatures, the frequency of the molecular rearrangement increases, accelerating the creep process. For fiber-reinforced polymers, nonetheless, the elastic deformation and viscous flow are delayed due to the presence of reinforcing fibers.

The predictions using the Findley law are included in Figure 7 as solid lines. As can be observed, the predictions using the parameters shown in Table 5 fit very well the experimental data. Interestingly, the Findley model provides precise predictions for all single-walled and double-walled rings, all winding angles, and conditions (ML, ML + WRT, ML + HW). The good correlation between experimental observations and the fitted values using Findley model can be verified through residual plots shown in Figure 8.

It is important to mention that two different effects are present, namely the reduction in stiffness due to the temperature and water uptake, and viscoelastic effects due to the temperature. These can be better assessed by comparing the static response of ML and ML + WRT. Figure 9 summarizes the creep stiffness results, defined by the applied force and the displacement at 240 h in each case, and a clear difference is observed between ML and ML + WRT conditioning (in green and blue, respectively). As aforementioned, this is strongly related to the static response of the rings and the reduction in stiffness is basically due to fiber/matrix weakening and the stiffness reduction of the epoxy resin. For the ML + HW, nevertheless, the viscoelastic response also plays an important role in ring displacement—a reduction of ca. 43% in stiffness is observed comparing ML and ML + HW responses. Furthermore, the role of the winding angle should be mentioned, and the closer the angle to the loading direction, the higher the creep stiffness and the lower the deflection.

In general, the post-creep rings have a similar ellipsoidal shape, independently of the stacking sequence and creep conditions (see Table 1). Representative post-creep specimens are shown in Figure 10. The experienced ovality in the ring cross-section is considered to be an irregularity and reduces the second moment of area of the cross-section. This induces higher stresses than the stress in a circular cross-section. Hence, ring ovalization may cause a detrimental effect on the residual properties [31].

It is also important to notice that only a small portion of the equilibrium trajectory of the creep specimens has a non-linear behavior, i.e., the creep curves are dominated by well-defined linear slopes. This, however, cannot be directly related to the creep regions, such as primary (strain-hardening) and secondary (constant-rate strain). The load-displacement behavior of such specimens can be non-linear since the ring (Figure 11a) starts becoming an ellipse (Figure 11b). For instance, the sample [±75] has shows 26% ovality, calculated as ($\frac{OD-ID}{ND}$), where OD, ID, and ND are outer, inner, and nominal diameters, respectively. The membrane force does not significantly change but the bending moment increases with displacement, mainly at the platens/ring contact and the side of the ring (minor and major axes of the ellipse formed when the ring is loaded—see Figure 11b), increasing the stress during creep.

Lisbôa et al. [32] demonstrated that the winding pattern plays a key role in FW composite rings under radial compressive loading, and the damage mechanisms can vary with the pattern. In agreement with that, Figure 12 shows a representative image of a post-creep ring highlighting the generated crack along the ring circumference. A similar “quasi-hoop” crack through the ring circumference along the +θ and −θ angles has been observed for all rings. Therefore, a decay in residual properties is expected given the permanent damage generated by the hygro-thermo-mechanical creep loading.

### 3.4. Residual Properties

The residual properties of the composite rings are presented in Figure 13 and Figure 14. Three tests are carried out: radial compression (Figure 13), axial compression (Figure 14), and hoop tensile strength (Figure 15).

Figure 13 depicts the residual properties of the rings in radial compression. In general, the results (Figure 13a) show similar trends, and the peak loads for UC, ML, and ML + WRT are within their deviation whereas significant differences in peak loads are observed for ML + HW conditioning. This decrease in load is essentially due to the deterioration process induced by both moisture and high temperature, influencing the load-carrying capacity of the rings [33]. The stiffness of the rings is shown in Figure 13b and is found to be influenced by both creep load and applied environmental conditions, especially for the ML + HW case. The change in stiffness is not substantial, in general, for the single-layer rings mainly because the thin walls have a limited amount of material to deteriorate. When a hoop layer is added on top of the single-layer, stiffness increases dramatically, and the later decrease in residual stiffness is more pronounced. For both [±60/±90] and [±75/±90] samples, the harsher the environment, the lower the residual radial compressive stiffness, with a reduction of 14% and 26%, respectively (residual stiffness of 364 N/mm and 521 N/mm, respectively).

Figure 13c shows the percentage ring deflection. These values are directly associated with the ring stiffness presented in Figure 13b. As the winding angle approaches the longitudinal axis and the diameter-to-thickness ratio increases, the load-bearing capacity decreases, and the cross-section changes from a circle to an ellipse [17]. When the UC specimens submitted to creep are compared, lower deflections are observed for the single-layer rings. Similar to stiffness results, significant changes are seen for the double-layer rings, and the ML + HW samples deformed significantly more than the others. Figure 13d presents the stiffness factor for all rings, which is a useful design parameter related to bending modulus and wall thickness (see Equation (Equation 5)), and also deflection [34]. The stiffness factor follows the same trend of the ring stiffness (Figure 13b), being higher for the rings with winding angles closer to ±90. More pronounced changes were observed for double-layer rings, in which ML + HW samples showed the lowest SF values and UC samples the highest.

Figure 14 presents residual properties for the rings under quasi-static axial compression. Regarding residual maximum load (Figure 14a), all single-layer cylinder showed similar values, i.e., residual peak load is not significantly affected by creep or hygro-thermo-mechanical conditionings. The same was found for the [±75/±90] ring. However, for the [±60/±90] ring, the maximum residual peak load follows UC > ML > ML + WRT > ML + HW. This implies that the ±60${}^{\circ }$ layers are more susceptible to damage than the ±75${}^{\circ }$ layers since the fibers are less aligned in the loading direction during creep, which makes the rings more dependent on the viscoelastic nature of the epoxy resin. Furthermore, ±60${}^{\circ }$ oriented composites tend to have more voids and resin-rich areas than ±75${}^{\circ }$ and ±90${}^{\circ }$ layers [23] and, as a consequence, they are more prone to water diffusion that harms the fiber/matrix interface [5]. When they are also at a higher temperature, damage at the interface is promoted.

Figure 15 shows residual hoop tensile load (Figure 15a) and strength (Figure 15b). Here, both residual peak loads and hoop tensile strength follow the same trend: UC > ML > ML + WRT > ML + HW. This trend is because, in this type of loading, the machined notches on the specimens yield stress concentration around the notches, making these samples more sensible to fiber/matrix interfacial changes. This is observed in Figure 15, in which mechanical loading combined with water or hot water intensifies the damage during creep and, therefore, the hoop tensile properties are more affected than radial and axial compressive ones.

## 4. Conclusions

The focus of this study was to design a creep test that allows combined hygrothermal conditioning and mechanical radial compression loading for filament-wound composite rings. The rings with one ([±60], [±75] and [±90]) or two layers ([±60/±90] and [±75/±90]) have been manufactured and tested. Three conditions have been employed: only mechanical loading, hygro-mechanical loading, and hygro-thermal-mechanical loading. The creep tests were performed at a constant loading corresponding to 25% of the maximum radial compressive load for 240 h, and residual properties were verified based on quasi-static testes, namely radial compression, axial compression, and hoop tension.

The rings under how water (HW) absorbed more water than at room temperature (WRT), in which the temperature leads to enhancement of absorbed water, whose phenomenon is known as thermal spiking. In addition, Fick’s law predicted very well the behavior of all specimens, but the [±60/±90] one, which showed a two-stage transient stage before reaching the pseudo-equilibrium. Regarding creep, a strong influence of the thermomechanical conditioning on ring deflection was observed. The rings presented higher deflection when under mechanical loading in hot water, followed by those under mechanical loading in water at room temperature, and those under mechanical loading only. A Findley model has been successfully employed, fitting very the experimental data for all rings. Moreover, the maximum reduction in residual properties reached 33% for radial compression, 18% for axial compression, and 14% for hoop tension, showing that the creep test generated a permanent damage in the rings.

In all, the results herein presented are very useful in understanding the effect of simultaneous loading and harsh environments on fiber-reinforced composite structures with off-axis layers.

## Figures and Tables

**Figure 1 polymers-13-00033-f001:**
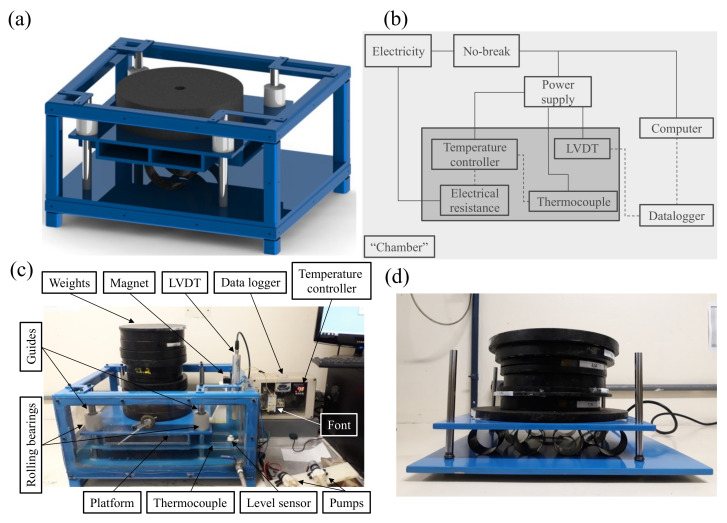
Creep testing equipment where (**a**) design of the creep testing equipment; (**b**) schematics of the whole system; (**c**) creep testing set up; and (**d**) specimens under creep loading.

**Figure 2 polymers-13-00033-f002:**
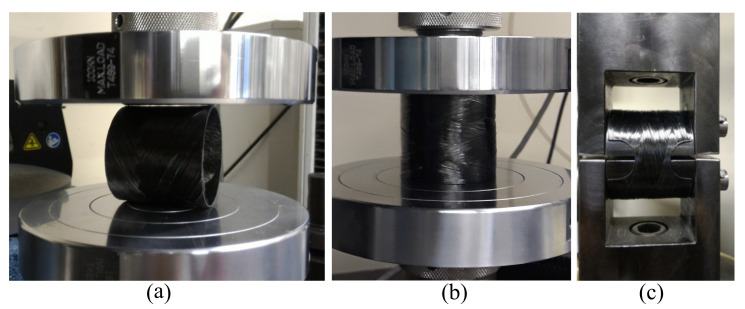
Quasi-static (**a**) radial compression, (**b**) axial compression and (**c**) hoop tensile tests.

**Figure 3 polymers-13-00033-f003:**
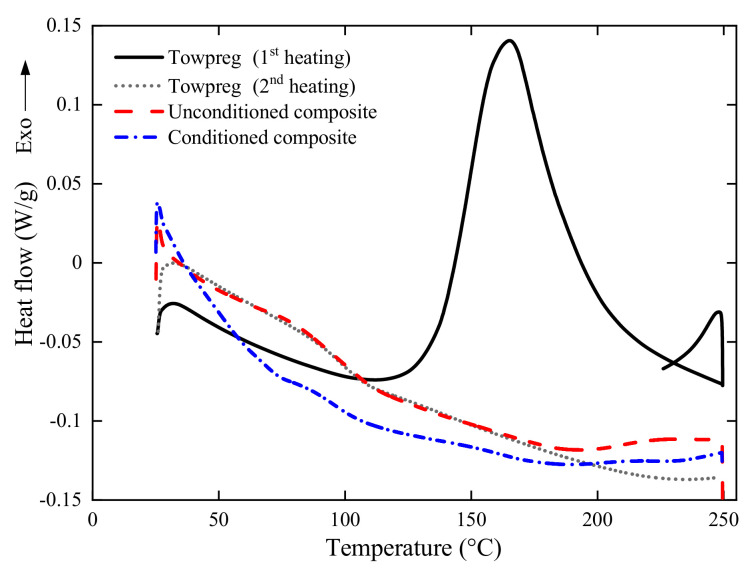
The DSC thermogram for the ring [±75] before and after creep exposure in water at 40 ${}^{\circ }$C for 240 h.

**Figure 4 polymers-13-00033-f004:**
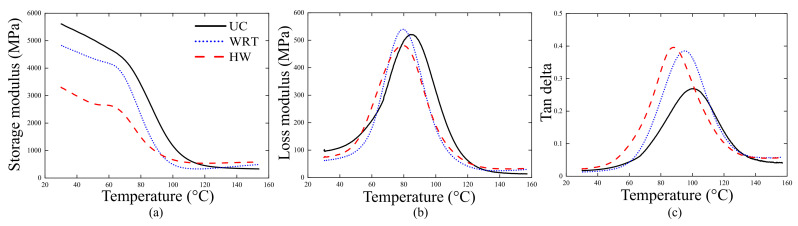
(**a**) Storage and (**b**) loss moduli, and (**c**) tan delta of the [±75] composite ring.

**Figure 5 polymers-13-00033-f005:**
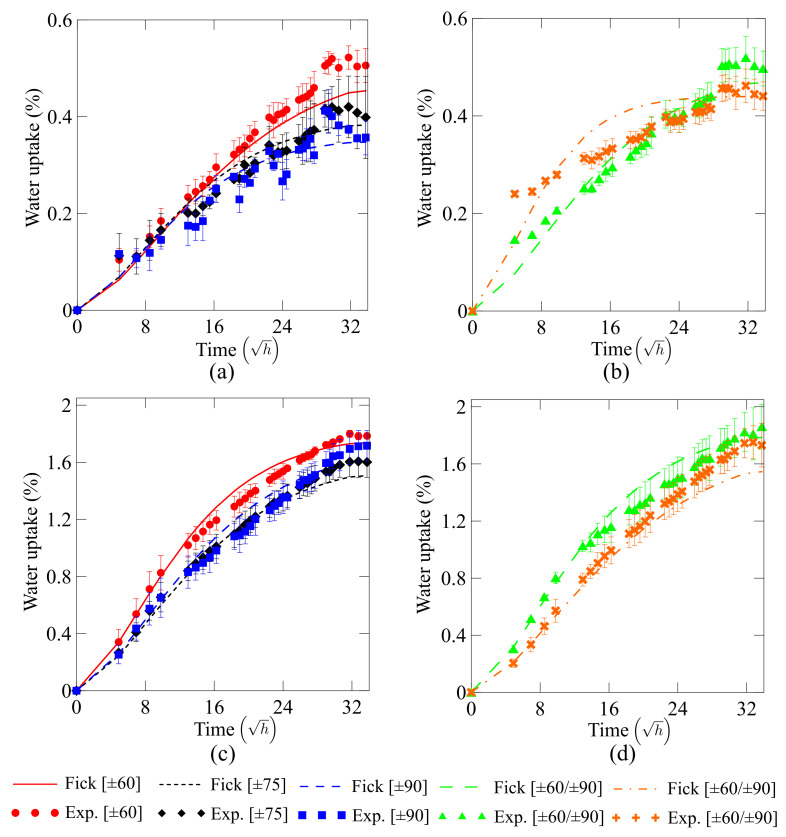
Water uptake for the composite rings under (**a**,**b**) water at room temperature (WRT) and (**c**,**d**) hot water (HW). Vertical bars represent the standard deviation of the measures. Figures are plotted with different axes values for better interpretation of the results.

**Figure 6 polymers-13-00033-f006:**
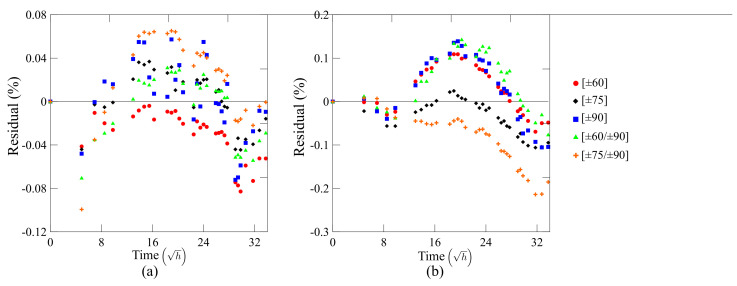
Residual plots comparing experimentally measured water uptake for the rings and predictions from Fick’s law for (**a**) water at room temperature (WRT) and (**b**) hot water (HW).

**Figure 7 polymers-13-00033-f007:**
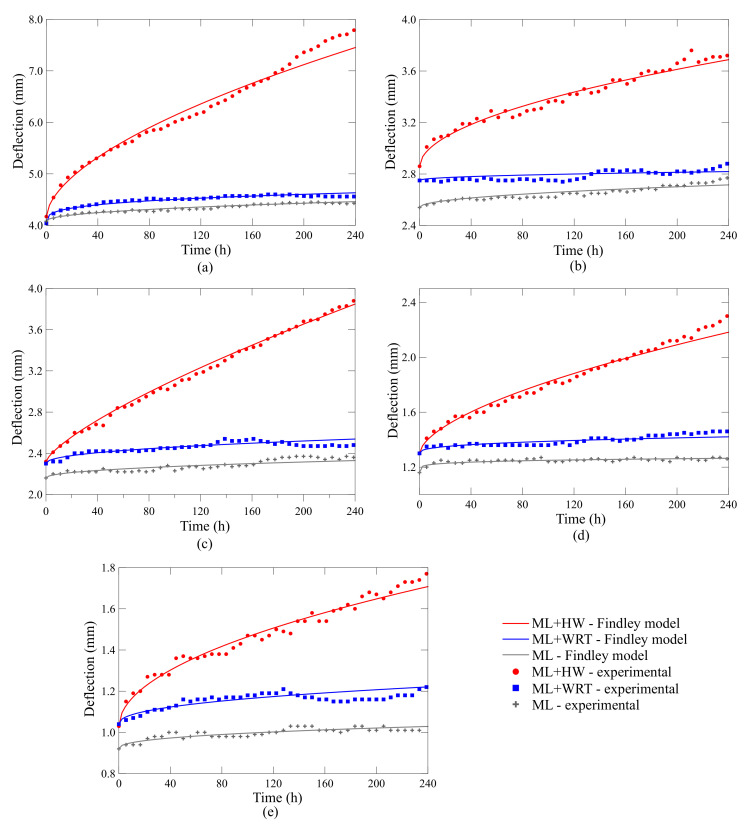
Creep of mechanical (ML), hygro-mechanical (ML + WRT) and hygro-thermal-mechanical (ML + HW) conditioned rings: (**a**) [±60], (**b**) [±75], (**c**) [±90], (**d**) [±60/±90], and (**e**) [±75/±90].

**Figure 8 polymers-13-00033-f008:**
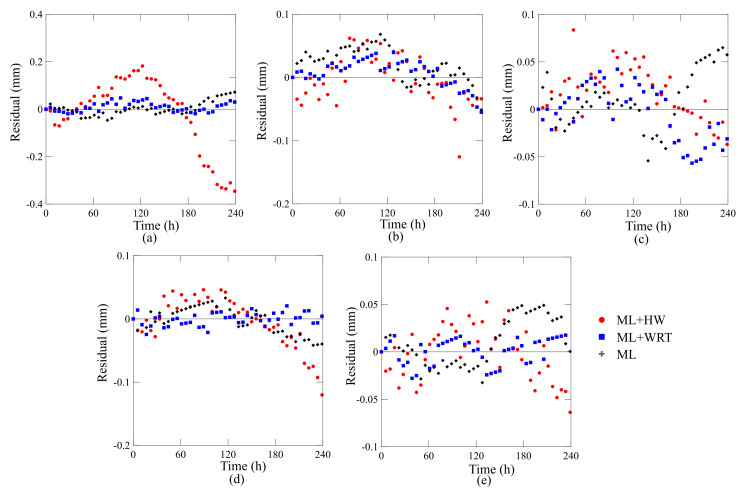
Residual plots of the Findley’s model with respect to the experimental data for the rings: (**a**) [±60], (**b**) [±75], (**c**) [±90], (**d**) [±60/±90], and (**e**) [±75/±90].

**Figure 9 polymers-13-00033-f009:**
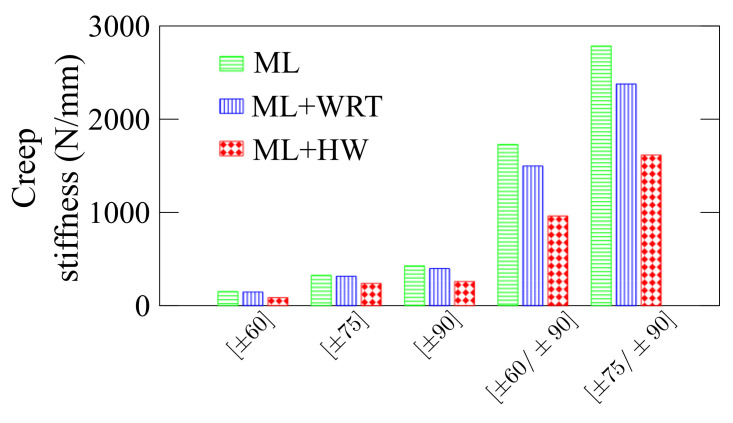
Creep stiffness for all composite rings. The stiffness is defined by the applied load and the measured displacement at 240 h.

**Figure 10 polymers-13-00033-f010:**
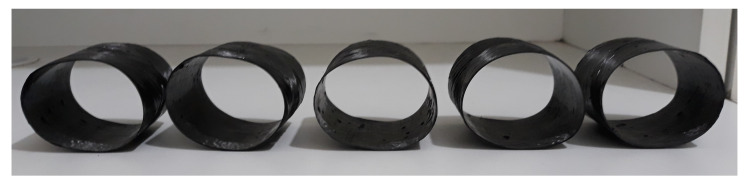
Representative ML + HW post-creep samples.

**Figure 11 polymers-13-00033-f011:**
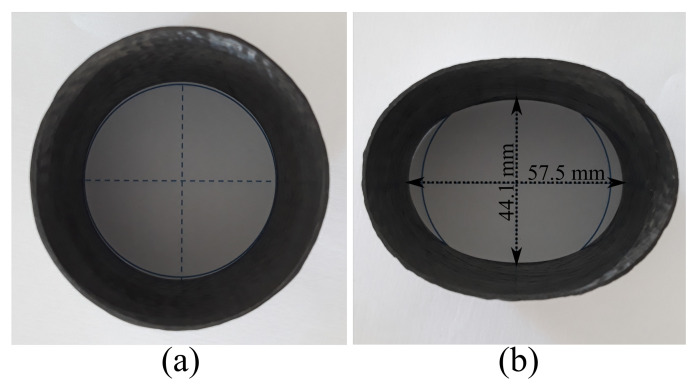
(**a**) Pre- and (**b**) post-creep representative ring.

**Figure 12 polymers-13-00033-f012:**
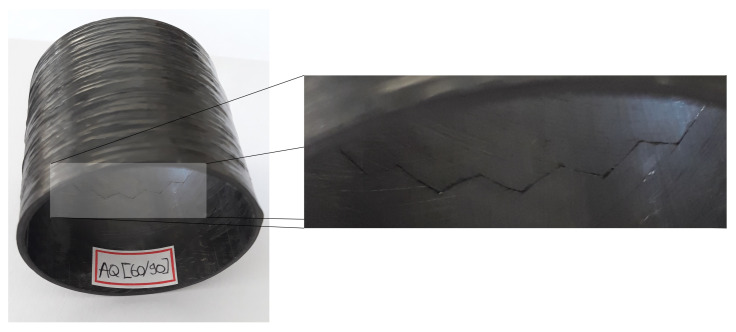
Post-creep [±60/±90] ring highlighting a crack at the cross-over region.

**Figure 13 polymers-13-00033-f013:**
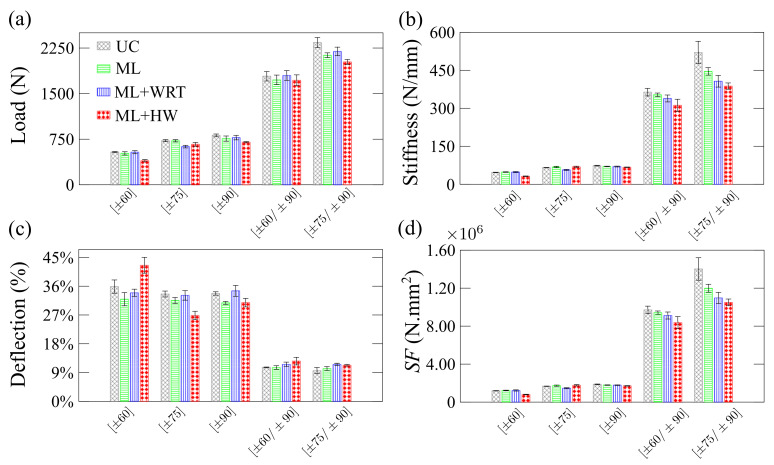
Residual properties of the rings under radial compression: (**a**) maximum load-carrying capacity, (**b**) stiffness (Equation (Equation 3)), (**c**) percentage deflection (Equation (Equation 4)), and (**d**) stiffness factor (Equation (Equation 5)).

**Figure 14 polymers-13-00033-f014:**
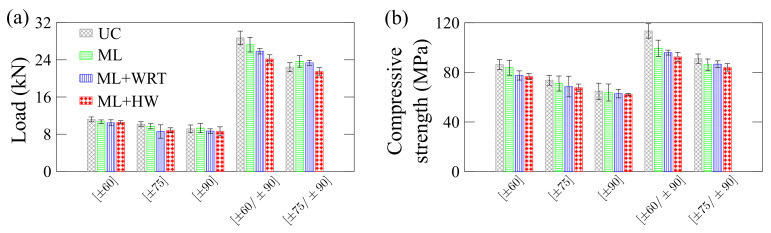
Residual properties of the rings under axial compression: (**a**) maximum supported load and (**b**) compressive strength.

**Figure 15 polymers-13-00033-f015:**
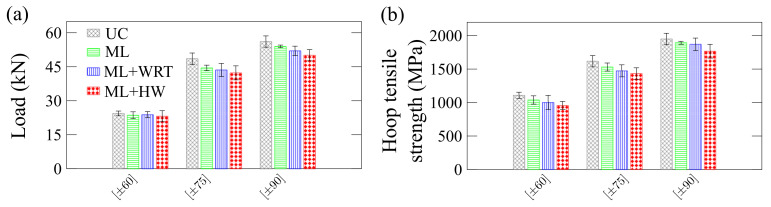
Residual properties of the rings under hoop tensile strength: (**a**) maximum loading and (**b**) strength.

**Table 1 polymers-13-00033-t001:** Adopted nomenclature for the samples.

*Nomenclature*	*Description*
**UC**	Baseline: **U**n**C**onditioned
**ML**	Creep in radial compression (**M**echanical **L**oading)
**ML + WRT**	Creep in radial compression (**M**echanical **L**oading) and **W**ater at **R**oom **T**emperature (23 ${}^{\circ }$C)
**ML + HW**	Creep in radial compression (**M**echanical **L**oading) and **H**ot **W**ater (40 ${}^{\circ }$C)

**Table 2 polymers-13-00033-t002:** Storage and loss moduli, and $T_{g}$ from DMA analysis.

Conditioning	$E_{max}^{\prime }$ (MPa)	$E_{max}^{\prime \prime }$ (MPa)	$T_{g,E^{\prime }}$ (${}^{\circ }$C)	$T_{g,E^{\prime \prime }}$ (${}^{\circ }$C)	$T_{g,\tan\delta }$ (${}^{\circ }$C)
ML	5601	502	79	86	101
ML + WRT	4822	540	75	79	95
ML + HW	3303	481	69	79	88

**Table 3 polymers-13-00033-t003:** Parameters used in the Fick’s model to predict the water uptake.

Laminate	Conditioning	$M_{\infty }$	*h* [mm]	*D* [mm^2^/h] (×10^−4^)
[±60]	WRT	0.506	0.72	1.03
	HW	1.784		1.92
[±75]	WRT	0.399	0.77	1.89
	HW	1.601		1.59
[±90]	WRT	0.357	0.70	1.83
	HW	1.716		1.29
[±60/±90]	WRT	0.497	1.54	9.63
	HW	1.861		7.47
[±75/±90]	WRT	0.441	1.69	28.3
	HW	1.731		5.58

**Table 4 polymers-13-00033-t004:** Mean thickness of the samples (50-mm long and 50.8 mm inner diameter on average) and applied creep load [23].

Laminate	Thickness [mm]	Applied Load [N]
[±60]	0.72 ± 0.016	135.0
[±75]	0.77 ± 0.006	181.5
[±90]	0.70 ± 0.010	203.0
[±60/±90]	1.54 ± 0.007	446.5
[±75/±90]	1.69 ± 0.017	585.0

**Table 5 polymers-13-00033-t005:** Parameters for the Findley model that best fit the creep response for all rings.

	[±60]			[±75]			[±90]		
	ML	ML + WRT	ML + HW	ML	ML + WRT	ML + HW	ML	ML + WRT	ML + HW
*d* _0_	4.08	4.04	4.17	2.54	2.75	2.86	2.16	2.30	2.32
*A* (×10^−2^)	3.028	12.84	13.30	1.243	0.547	4.712	1.301	1.977	2.548
*n*	0.462	0.279	0.585	0.483	0.462	0.523	0.468	0.454	0.747
	[±60/±90]			[±75/±90]					
	ML	ML + WRT	ML + HW	ML	ML + WRT	ML + HW			
*d* _0_	1.16	1.30	1.30	0.92	1.04	1.03				
*A* (×10^−2^)	4.000	1.694	3.292	1.183	1.676	4.165				
*n*	0.175	0.358	0.600	0.405	0.434	0.509			

## Data Availability

All the experimental data herein presented are made available upon request to the corresponding author.

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
