# Peer review of "Creep and Residual Properties of Filament-Wound Composite Rings under Radial Compression in Harsh Environments"

_polymers, 2020, doi:10.3390/polym13010033_

Round 1
Reviewer 1 Report
Review
- Line 203: Initial observation indicates that Fick’s law fits the experimental data very well in both cases (water at room temperature and hot water).
- Figure 5. Water uptake for the composite rings under (a) water at room temperature (WRT) and (b) hot water (HW).
- Line 215: Moreover, as one can see in Figure 5, the temperature strongly influences water uptake.
Comment:
Apparently, The Fick model fittings apparently are not appropriate ones. Some statistical parameters and residuals plots would be helpful.
No Fick model terms (the mass at a quasi-equilibrium state, the diffusion coefficient) are given so that we can conclude on the thermal activation of water uptake.
Of course, the water uptake (in %) would be different for cold and hot water, but the scaling in Fig5a is too large to see the difference in uptake for different windings. It would be also much to give times x-axis in logarithmic scale! Different coloring of the Vertical bars would also be appreciated.
- Figure 6. Creep of mechanical (ML, in black), hygro-mechanical (ML+WRT, in blue) and hygrothermal-mechanical (ML-HW, in red)
Comment: In Fig.6 inserts were given for the short time. It would be more appropriate to give times x-axis in logarithmic scale, without need for the inserts. Deflection scale is also given extended () so that differences are not apparent).
Vertical bars that show the standard deviation of the (5?) measures are not given!
Would be more informative when the creep of under different mechanical, hygro-mechanical and hygrothermal-mechanical conditions for various windings are given in the same Figure?
Logarithmic, exponential, power functions or some non-linear creep model may have been tried to describe the creep-time relation with few fitting parameters.
- Figure 4. (a) Storage and (b) loss moduli, and (c) tan delta of the [±75] composite ring.
Comment: The DMA test should have been performed at least for some other rings, at limits, [±60], [±90] or some other rings…
- Line 239: In general, the post-creep rings have a similar ellipsoidal shape, independently of the stacking sequence and creep conditions.
Comment: The ovality should have been determined! at least as ratio of ellipsoid axis. see Figure 9.
- Line 254: Figure 10 shows a representative image of a post-creep ring
Comment: Have You observed other damages on other windings?

Author Response
Dear Reviewer,
all comments were considered in the revised version of the manuscript and the responses to the reviewer are presented here. The modifications are highlighted in blue in the revised manuscript. The authors would like to thank you for the important recommendations, which surely improved the readability and technical rigor of the manuscript.
Reviewer #1:
- Line 203: Initial observation indicates that Fick’s law fits the experimental data very well in both cases (water at room temperature and hot water).
- Figure 5. Water uptake for the composite rings under (a) water at room temperature (WRT) and (b) hot water (HW).
- Line 215:Moreover, as one can see in Figure 5, the temperature strongly influences water uptake.
Comment: Apparently, The Fick model fittings apparently are not appropriate ones. Some statistical parameters and residuals plots would be helpful. No Fick model terms (the mass at a quasi-equilibrium state, the diffusion coefficient) are given so that we can conclude on the thermal activation of water uptake.
We agree. We added a Table with the Fick parameters for all rings, and we added residual plots as requested. Also, although the Fick’s model does not precisely predict the whole period, it generates overall excellent results. The observed deviation is justified considering that the analytical model does not consider the winding pattern formed during manufacturing, which is only possible via a numerical approach.
Of course, the water uptake (in %) would be different for cold and hot water, but the scaling in Fig5a is too large to see the difference in uptake for different windings. It would be also much to give times x-axis in logarithmic scale! Different coloring of the Vertical bars would also be appreciated.
We agree. We replotted Fig5a following the maximum and minimum values in the axes and vertical using bars with the same color of the respective composite sample for consistency. And we separated them into 4 graphs to better visualize the results (see Figure 5).
- Figure 6. Creep of mechanical (ML, in black), hygro-mechanical (ML+WRT, in blue) and hygrothermal-mechanical (ML-HW, in red)
Comment: In Fig.6 inserts were given for the short time. It would be more appropriate to give times x-axis in logarithmic scale, without need for the inserts. Deflection scale is also given extended () so that differences are not apparent).
We replotted all creep graphs without the inserts and maximized the frame for each composite sample. We tried plotting using log scale in the x-axis, but the plots became too crowded for longer times, so this was not done to increase readability. Please check Figure 7.
Vertical bars that show the standard deviation of the (5?) measures are not given!
Five specimens of each composite sample were simultaneously subjected to creep in the apparatus to ensure that the 5 specimens underwent exactly the same creep loading, creep strain and environmental conditioning. Therefore, there are no standard deviations (vertical bars) for creep measurements.
Would be more informative when the creep of under different mechanical, hygro-mechanical and hygrothermal-mechanical conditions for various windings are given in the same Figure?
Logarithmic, exponential, power functions or some non-linear creep model may have been tried to describe the creep-time relation with few fitting parameters.
Thanks for this valuable comment. We decided to add a Findley-like power law model to fit the experimental data. All parameters that best fitted the experiments are shown in Table 5. Please check also Figure 8.
- Figure 4. (a) Storage and (b) loss moduli, and (c) tan delta of the [±75] composite ring.
Comment: The DMA test should have been performed at least for some other rings, at limits, [±60], [±90] or some other rings…
In previous works of the group (Ref [5]), we observed that the Tg is not dependent on the winding angle, which makes sense since the material system is the same. Therefore, we chose only one sample to determine the Tg.
- Line 239: In general, the post-creep rings have a similar ellipsoidal shape, independently of the stacking sequence and creep conditions.
Comment: The ovality should have been determined! at least as ratio of ellipsoid axis. see Figure 9.
We agree with the Reviewer. We now reported ovality data (see Figure 9).
- Line 254: Figure 10 shows a representative image of a post-creep ring
Comment: Have You observed other damages on other windings?
Yes, we observed a similar crack in all windings. We modified the text accordingly.
Best regards,
THE AUTHORS.
Reviewer 2 Report
Journal: Polymers (ISSN 2073-4360)
Manuscript ID: polymers-1045821
Creep and residual properties of filament wound composite rings under radial compression in harsh environment.
About the manuscript, the introduction should be more detailed in the problem to resolve and the literature. Experimental conditions can be more detailed and the used conditions more precise and justified. The English is good, just few typos was found.
One typo in manuscript is the lack of reference in the shown equations. Authors should reference the parameters introduced in the manuscript. Non-specialists must be able to verify the origin of the information.
The parameters used must be justified. Why these dimensions of the samples? The temperature of the water?
The cure temperature of the cylinders was optimized? Authors can add one reference.
The thickness and diameter influence the overall mechanical measurements? Although, what is the carbon contents in resin? And can influence the water absorption or even the mechanical properties?
“Thus, comparing the DSC curves of the towpreg (2nd heating) and the unconditioned composite, an adequate degree of curing of the epoxy resin is found based on the coinciding curves and absence of an exothermic peak.” – Can determine the “adequate degree?”
The Figure must be like the Figure 5b, maximizing the area of the graphic to improved observation of the results.
Why the water uptake results of the +-60/90 results are far away from the theoretical fit, compared with other results? Higher uptake for smaller times, and the inverse for 1000 h, compared with other results.
In Figure 6, the deflection with time increase with time for all ML+HW samples. But these increases is higher for lower diameters. Why this behavior?
Author Response
Dear Reviewer,
all comments were considered in the revised version of the manuscript and the responses to the reviewer are presented here. The modifications are highlighted in blue in the revised manuscript. The authors would like to thank you for the important recommendations, which surely improved the readability and technical rigor of the manuscript.
Reviewer #2
Creep and residual properties of filament wound composite rings under radial compression in harsh environment.
About the manuscript, the introduction should be more detailed in the problem to resolve and the literature. Experimental conditions can be more detailed and the used conditions more precise and justified. The English is good, just few typos was found.
We agree with the reviewer. We modified the Introduction (last two paragraphs) aiming at detailing more explicitly the problem.
One typo in manuscript is the lack of reference in the shown equations. Authors should reference the parameters introduced in the manuscript. Non-specialists must be able to verify the origin of the information.
References to all Equations have been added.
The parameters used must be justified. Why these dimensions of the samples? The temperature of the water?
Dimensions of the samples were chosen based on a previous study of the group (Carbon/epoxy filament wound composite drive shafts under torsion and compression). And the temperature was selected following the European standard EN 1227:1998, used to determine long-term ultimate relative ring deflection under wet conditions. We included that information in the manuscript.
The cure temperature of the cylinders was optimized? Authors can add one reference.
We followed the manufacturer`s recommendations to cure the towpreg. In addition, curing was studied in several previous studies of the group, which are now referred to in the manuscript (Section 2.1).
The thickness and diameter influence the overall mechanical measurements? Although, what is the carbon contents in resin? And can influence the water absorption or even the mechanical properties?
The geometric characteristics of the rings influence both the mechanical and physical (water uptake) properties. The former can be observed in Figures 11-13, where the thickness (and hence, diameter) is considered in the strengths calculations. However, the sample thickness does not significantly affect water uptake. The fiber volume fraction is 72% - this information has now been added in Section 2.1.
“Thus, comparing the DSC curves of the towpreg (2nd heating) and the unconditioned composite, an adequate degree of curing of the epoxy resin is found based on the coinciding curves and absence of an exothermic peak.” – Can determine the “adequate degree?”
We attribute “adequate curing” to the composites considering the absence of an exothermic peak in the second heating. The text was modified to clarify that.
The Figure must be like the Figure 5b, maximizing the area of the graphic to improved observation of the results.
We agree. The figure was modified.
Why the water uptake results of the +-60/90 results are far away from the theoretical fit, compared with other results? Higher uptake for smaller times, and the inverse for 1000 h, compared with other results.
This sample showed a distinct behavior, with a two-transient-stage, i.e. after the transient state in water uptake, a second transient stage took place at a more gradual weight uptake. Only after that the pseudo-equilibrium stage (and therefore the maximum uptake) is reached. This behavior can only be captured by a model that incorporates structural relaxation in the Fick’s law. We’ll keep that possibility open for future work.
In Figure 6, the deflection with time increase with time for all ML+HW samples. But these increases is higher for lower diameters. Why this behavior?
The inner diameter is the same for all rings, but the outer diameter slightly vary given the particular thicknesses. Nonetheless, deflection for ML+HW samples increases with time for thinner specimens because they have less membrane strength to support the creep loading under hot water. Besides, these specimens are more matrix-dependent, making them more susceptible to deflection than thicker specimens.
Best regards,
THE AUTHORS.